# Grand multiparity and its associated factors in Zambia: Evidence from the 2018 Zambia Demographic and Health Survey

Peter Mumba[1]*, Emmanuel Musonda[1], Mwangala Mulasinkwanda[1], Jessy Mutale Nkonde[2], Vanessa Moonga[3], Stephen Jere[4], Samson Shumba[2]

1 Department of Demography, Population Sciences, Monitoring and Evaluation School of Humanities and Social Sciences, University of Zambia, Lusaka, Zambia, 2 Department of Epidemiology and Biostatistics, School of Public Health, University of Zambia, Lusaka, Zambia, 3 Department of Economics, School of Business, University of Lusaka, Lusaka, Zambia, 4 Department of Monitoring, Evaluation and Learning, Avencion Limited, Lusaka, Zambia

* petmumba50@gmail.com

## Abstract

Grand multiparity, defined as having five or more births, is a significant public health problem, especially in low resource settings where fertility rates are generally higher than in high-resource settings due to socio-economic factors such as poverty and low levels of women's education, as well as disparities in access to modern contraceptives and advanced medical care. The study aimed to investigate the factors associated with grand multiparity in Zambia. The study analysed data from the 2018 Zambia Demographic and Health Survey (ZDHS) using a weighted sample of 13,683 women aged 15–49 years. Univariate analysis was used to present percent distributions of the selected background characteristics of women. Bivariate analysis was used to examine the association between selected background characteristics and grand multiparity. Multivariable logistic regression was then performed to identify factors of grand multiparity among women aged 15–49 years in Zambia. The prevalence of grand multiparity among women in Zambia was estimated at 24%. Women aged 25–29 years, 30–34 years, and 35 years or older had significantly higher odds of grand multiparity compared to women aged 15–24 years, with the highest odds observed in women aged 35 years or older (aOR=56.31, 95% CI: 11.626–272.780). Women with a first birth at age 25 years or older were 86% less likely to experience grand multiparity (aOR=0.14, 95% CI: 0.084–0.238). Higher education and wealth were protective factors, with women who attained secondary education (aOR=0.28, 95% CI: 0.182–0.418) and higher education (aOR=0.09, 95% CI: 0.046–0.174) having reduced odds of grand multiparity. Women from richer households were also less likely to experience grand multiparity (aOR=0.48, 95% CI: 0.337–0.701). Rural residence (aOR=1.46, 95% CI: 1.111–1.911) and contraceptive use (aOR=1.29, 95% CI: 1.074–1.545) were associated with increased odds of grand multiparity. Women

which permits unrestricted use, distribution, and reproduction in any medium, provided the original author and source are credited.

**Data availability statement:** Data for this study were obtained from the Zambia Demographic and Health Survey (ZDHS), which is part of the Demographic and Health Surveys (DHS) Program. The data were originally accessed through the DHS Program website (https://dhsprogram.com/). However, at the time of publication, the specific dataset used in this study was no longer publicly accessible through the website.

**Funding:** The authors received no specific funding for this work.

**Competing interests:** The authors have declared that no competing interests exist.

with shorter birth intervals (less than 24 months) were almost three times more likely to be grand multiparous (aOR=2.74, 95% CI: 1.741–4.326). The findings underscore the complex interplay of socio-demographic factors in shaping reproductive patterns in Zambia. Strategies to reduce grand multiparity should consider promoting delayed marriage, increasing access to education and addressing rural-urban disparities in healthcare access and family planning services.

## Background

Grand multiparity remains a significant concern in obstetrics due to its association with various medical and obstetric complications [1,2]. In societies where large families are highly valued, it is essential to emphasize the benefits of family planning and ensure comprehensive antenatal care [3]. The term "grand multiparity" was first introduced by Solomon in 1934, who referred to it as "dangerous multipara" [4]. The International Federation of Gynecology and Obstetrics (1993) defined grand multiparity as having five to nine viable pregnancies. Women who have undergone 10 or more deliveries are referred to as "great grand multipara" [5].

Research has shown that grand multiparity is associated with numerous maternal and perinatal complications, including pre-labour rupture of membranes, stillbirth, anemia, perinatal mortality, and preterm birth [6–8]. Fertility rates are generally higher in low resource settings compared to high resource settings, where advanced medical technology and accessible modern contraceptives are available [9]. This makes grand multiparity a serious global public health problem, particularly in low resource settings [9]. While the global fertility rate declined from 3.2 to 2.5 live births per woman between 1990 and 2019, in sub-Saharan Africa (SSA), it is projected to take 34 years, from 1995 to 2029, for fertility to decrease from 6 to 4 live births per woman [9].

Several factors are associated with grand multiparity, including illiteracy, non-use of contraceptives, short birth intervals, advanced maternal age, low education levels, and polygamous unions [10–13]. While research has primarily focused on predicting grand multiparity and its obstetric challenges, most studies have concentrated on the impact of grand multiparity on pregnancy outcomes [6,7,11,14–17].

The prevalence of grand multiparity in SSA countries ranges between 17% and 33% [18]. This high prevalence is often attributed to lower standards of living, inadequate healthcare resources, and limited use of modern contraceptives. In contrast, high-resource settings outside SSA typically report a much lower prevalence, ranging from 3% to 4% [18].

However, few studies have explored the determinants of grand multiparity using nationally representative samples, and to our knowledge, no such study has been conducted in Zambia. Most of the studies that have been done on grand multiparity have used hospital-based data. The findings from this research will provide critical

insights for the Ministry of Health (MoH) and implementing partners, helping to develop effective strategies to reduce the prevalence and associated risks of grand multiparity.

## Methods

### Study design

This study is a secondary data analysis of the existing national level data from the Zambia Demographic and Health Survey (ZDHS) program. The ZDHS is a nationally representative household survey conducted by the Zambia Statistics Agency with support from global partners including the ICF, Department for International Development (DFID) and United States Agency for International Development (USAID). The DHS uses two-stage sampling to select enumeration areas (EAs) in the first stage and households in the second stage. The nature of DHS data allows for comparisons between indicators over time, thus allowing monitoring changes in the indicators of interest in different geographical areas. Women aged 15–49 years from selected households who consented to take part in the study were enrolled in the survey. For this study, all relevant variables were extracted from the women individual recode (IR).

### Study variables

Table 1 presents the outcome variable and independent variables used in the study. The outcome variable, grand multiparity, was derived from the total number of children a woman has ever had in her lifetime. Grand multiparity was coded as having five or more children (1), while women with 0–4 children were coded as controls (0). For the univariate analysis, women aged 15–19 years and 20–24 years were analyzed as separate categories. However, in the bivariate and multivariate analyses, they were combined into a single category because none of the women aged below 20 were grand

**Table 1. Dependent and independent variables.**

| Dependent variable | |
|---|---|
| Grand Multiparity | 5 or more children (1), (0–4 children) controls (0) |
| **Independent variables** | |
| Age | 15-19, 20–24, 25–29, 30–34 and 35+ |
| Age at first birth | <20, 20-24, 25+ |
| Age at first sex | <20, 20-24, 25+ |
| Age at first marriage | <20, 20-24, 25+ |
| Marital duration | 0-4, 5-9, 10-19, 20+ |
| Maternal education | No education, Primary, Secondary, Higher |
| Wealth index | Poor, Middle, Rich |
| Employment status | No, Yes |
| Husband/partner education level | None, Primary, Secondary/Higher |
| Religion | Catholic, Protestant, Muslim, Other |
| Region | Central, Copperbelt, Eastern, Luapula, Lusaka, Muchinga, Northern, North-western, Southern, Western |
| Residence | Urban, Rural |
| Access to media | No, Yes |
| Marital status | Single, Married, Widowed/Separated/Divorced |
| Polygamous marriage | No, Yes |
| Contraceptive use | No, Yes |
| Previous birth interval | <24 months, 24 + months |
| Access to health care | Big problem, Not a problem |

multiparous. Wealth quintiles were categorized into three groups, as done in other studies. [13,19]. Wealth has been categorized into three groups (poor, middle, rich) by combining the "poorest" and "poorer" categories, and the "richer" and "richest" categories, to ensure sufficient sample sizes for robust analysis and to align with the methodologies used in previous studies [13,19]. The variables, getting permission to go for treatment, getting money needed for treatment, distance to a health facility, having to take transport, and not wanting to go alone, were combined to form a composite variable representing access to healthcare, as each reflects a potential barrier to obtaining medical services [20].

### Statistical analysis

The statistical analysis for this study was conducted using Stata version 15 and involved three levels of analysis: univariate, bivariate, and multivariable. At the univariate level, descriptive statistics were generated to present the sample characteristics as percentages, highlighting the distribution of respondents across the selected variables. At the bivariate level, cross-tabulation was performed, and Pearson's chi-squared test was used to examine the association between grand multiparity and the independent variables.

For the multivariable analysis, binary logistic regression was employed to identify factors associated with grand multiparity in Zambia. To assess multicollinearity among the independent variables, the Variance Inflation Factor (VIF) was calculated, and marital status was found to be collinear and was therefore dropped from the model. After its removal, no further collinearity was detected among the remaining variables, all of which had a VIF of less than 5 as shown in S1 Table. Only variables that were significant at the 10% level were included in the final model. The goodness of fit for the model was assessed using the Hosmer-Lemeshow test, which yielded a p-value greater than 0.05, confirming that the model provided a good fit for the data. We sought to improve the generalizability of our results by addressing discrepancies introduced during the sampling process through the application of weights to the observations.

### Ethics

The study was based on the secondary ZDHS data, which is publicly available on (https://dhsprogram.com/). All data used did not contain any identifying information. The original ZDHS 2018 Biomarker and survey protocols were approved by Tropical Disease and Research Center (TDRC) and the Research Ethics Review Board of the Centers for Disease Control and Prevention (CDC). Informed consent was obtained from all the women who participated in the study and were 18 years or older. For women who were below the age of 18 years, their guardian or parent consented on their behalf. All the data from the DHS dataset are anonymized, there are no personal identifiers to link the information back to the respondent.

### Results

Table 2 presents the background characteristics of women aged 15–49 years in Zambia. The results show that 28% of women were aged between 35–49 years. The findings also reveal that more than two-thirds (68%) of the women had their first childbirth before the age of 20 years, and the majority (91%) initiated sexual activity before turning 20 years. Additionally, 68% of the women were married before reaching the age of 20 years, and nearly half (41%) had been married for over 20 years. In terms of education, about 42% of the women had completed primary education.

Table 2 further shows that 47% of the women were from rich wealth quintiles. About 45% of the women were employed. Most of the women identified as protestants (82%). About 53% of the women resided in rural areas, and 45% had access to media. The results also revealed that more than half (55%) of the women were married.

### Prevalence of grand multiparity of women aged 15–49 years

The study findings indicate that 24% of the women in Zambia had experienced grand multiparity as indicated in Fig 1. Further, results show a strong association between a woman's age and the likelihood of grand multiparity. Women aged 35

**Table 2. Percent distribution of women aged 15-49 by background characteristics, 2018 DHS, Zambia. N = 13683.**

| Covariates | Count (weighted) | Percent (weighted) |
|---|---|---|
| **Age** | | |
| 15-19 | 3000 | 21.9 |
| 20-24 | 2733 | 20.0 |
| 25-29 | 2237 | 16.4 |
| 30-34 | 1862 | 13.6 |
| 35+ | 3850 | 28.1 |
| **Age at first birth** | | |
| <20 | 6998 | 67.8 |
| 20-34 | 2685 | 26.0 |
| 25+ | 635 | 6.2 |
| **Age at first sex** | | |
| <20 | 12398 | 90.6 |
| 20-24 | 1150 | 8.4 |
| 25+ | 140 | 1.0 |
| **Age at first marriage** | | |
| <20 | 6375 | 67.7 |
| 20-24 | 2197 | 23.3 |
| 25+ | 838 | 8.9 |
| **Marital duration** | | |
| 0-4 | 1910 | 20.3 |
| 5-9 | 1991 | 21.2 |
| 10-19 | 1654 | 17.6 |
| 20+ | 3856 | 41.0 |
| **Maternal education** | | |
| No education | 1054 | 7.7 |
| Primary | 6059 | 44.3 |
| Secondary | 5816 | 42.5 |
| Higher | 755 | 5.5 |
| **Wealth index** | | |
| Poor | 4828 | 35.3 |
| Middle | 2477 | 18.1 |
| Rich | 6377 | 46.6 |
| **Employment status** | | |
| No | 7519 | 55.0 |
| Yes | 6164 | 45.0 |
| **Husband/partner education level** | | |
| No education | 425 | 5.8 |
| Primary | 6256 | 84.9 |
| Secondary | 690 | 9.4 |
| **Religion** | | |
| Catholic | 2354 | 17.2 |
| Protestant | 11098 | 81.1 |
| Muslim | 64 | 0.5 |
| Other | 167 | 1.2 |
| **Region** | | |

*(Continued)*

**Table 2.** (Continued)

| Covariates | Count (weighted) | Percent (weighted) |
|---|---|---|
| Central | 1165 | 8.5 |
| Copperbelt | 2200 | 16.1 |
| Eastern | 1605 | 11.7 |
| Luapula | 1071 | 7.8 |
| Lusaka | 2733 | 20.0 |
| Muchinga | 754 | 5.5 |
| Northen | 1054 | 7.7 |
| North-western | 718 | 5.2 |
| Southern | 1574 | 11.5 |
| Western | 808 | 5.9 |
| **Residence** | | |
| Urban | 6374 | 46.6 |
| Rural | 7309 | 53.4 |
| **Access to media** | | |
| No | 7560 | 55.3 |
| Yes | 6123 | 44.8 |
| **Marital status** | | |
| Single | 4272 | 31.2 |
| Married | 7648 | 55.9 |
| Divorced/Widowed/Separated | 1762 | 12.9 |
| **Polygamous marriage** | | |
| No | 6718 | 88.8 |
| Yes | 848 | 11.2 |
| **Contraceptive use** | | |
| No | 8844 | 64.6 |
| Yes | 4839 | 35.4 |
| **Previous birth interval** | | |
| 24 months or more | 6969 | 87.2 |
| Less than 24 months | 1025 | 12.8 |
| **Access to health care** | | |
| Big problem | 187 | 1.4 |
| Not a big problem | 13496 | 98.6 |

years or older had a significantly higher prevalence of grand multiparity (64%) compared to younger women. Women who began childbearing at an early age faced a greater risk of grand multiparity than women who delayed their first childbirth. The timing of marriage also influenced the likelihood of grand multiparity, with 40% of women married before the age of 20 years being grand multipara. Additionally, a marital duration of 20 years or more significantly increased the likelihood of grand multiparity, with over two-thirds (69%) of women in this category experiencing grand multiparity.

The results also show that women with no formal education had a higher prevalence of grand multiparity (52%) compared to women with higher education (4%). Wealth status also had a significant association; women from poor wealth quintile had a higher prevalence of grand multiparity (34%) compared to women from rich quintile (14%). Employment status also emerged as a significant factor, with 31% of employed women experiencing grand multiparity. Furthermore,

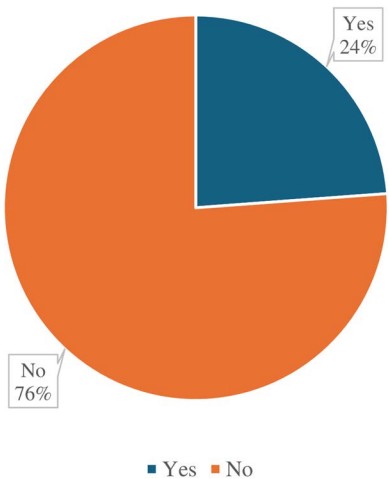

**Fig 1. Prevalence of grand multiparity in Zambia among childbearing women aged (15-49).**

rural residence (32%) was associated with a higher likelihood of grand multiparity compared to urban residence (15%), as shown in Table 3.

## Factors associated with grand multiparity among women aged 15–49 years

We applied a multivariable binary logistic regression to identify factors associated with grand multiparity among women aged 15–49 years in Zambia. The results show a strong, positive association between increasing age and grand multiparity. Women aged 25–29 years had significantly higher odds of grand multiparity compared to women aged 15–24 years (aOR=4.86, 95% CI: 3.450–73.645). This likelihood increased further for women aged 30–34 years (aOR=15.94, 95% CI: 1.344–26.087) and was highest among women aged 35 years or older (aOR=56.31, 95% CI: 11.626–272.780). Age at first birth was inversely associated with grand multiparity. Women who had their first birth between ages 20–24 years were 60% less likely to be grand multiparous (aOR=0.40, 95% CI: 0.327–0.507), while women whose first birth occurred at age 25 years or older were 86% less likely to be grand multiparous (aOR=0.14, 95% CI: 0.084–0.238) compared to women who gave birth before the age of 20 years.

In terms of age at first marriage, marrying at 25 years or older was associated with higher odds of grand multiparity (aOR=1.79, 95% CI: 1.075–2.969) compared to marrying before the age of 20 years. Women married for 10–19 years had more than five times the odds of grand multiparity (aOR=5.31, 95% CI: 2.295–12.325), and women married for 20 years or more had 14 times the odds grand multiparity (aOR=14.10, 95% CI: 5.854–33.957) compared to women married for less than 5 years as shown in Table 4.

Women with secondary education were 72% less likely to be grand multiparous (aOR=0.28, 95% CI: 0.182–0.418), and women with higher education were91% less likely to be grand multiparous (aOR=0.09, 95% CI: 0.046–0.174) compared to women with no education. Women from middle income quintile households were 33% less likely to be grand multiparous compared to women from poor quintile households (aOR=0.67, 95% CI: 0.507–0.897). Further, women from rich quintile household were 52% less likely to be grand multiparous compared to women from poor quintile households (aOR=0.48, 95% CI: 0.337–0.701). Women residing in rural areas were 46% more likely to be grand multiparious compared to women residing in urban areas (aOR=1.46, 95% CI: 1.111–1.911). The results show that women using contraceptives were 29% more likely to experience grand multiparity compared to women who were not using contraceptives (aOR=1.29, 95% CI: 1.074–1.545).

**Table 3. Bivariate analysis of women's parity (zero to low parity (2-4) and grand multiparity) by background characteristics, Zambia. N=13683.**

| Covariates | Zero to low parity (0–4 children) (N=10,423) | Grand multipara (5 or more children) (N=3,260) | (p-value) |
|---|---|---|---|
| | Percent (CI) | Percent (CI) | |
| **Age** | | | |
| 15-24 | 99.7 [99.0–99.9] | 0.3 [0.0–0.1] | <0.0001 |
| 25-29 | 92.2 [90.6–93.4] | 7.9 [6.6–9.3] | |
| 30-34 | 67.8 [65.0–70.4] | 32.2 [29.6–35.0] | |
| 35+ | 35.9 [33.6–38.3] | 64.1 [61.7–66.4] | |
| **Age at first birth** | | | |
| <20 | 64.3 [63.0–65.6] | 35.7 [34.4–37.0] | <0.0001 |
| 20-34 | 74.9 [72.9–76.9] | 25.1 [23.1–27.2] | |
| 25+ | 86.0 [82.3–89.0] | 14.0 [11.0–17.6] | |
| **Age at first sex** | | | |
| <20 | 75.2 [74.1–76.2] | 24.8 [23.8–25.9] | <0.0001 |
| 20-24 | 85.0 [81.9–87.6] | 15.0 [12.4–18.1] | |
| 35+ | 92.1 [81.8–96.8] | 7.9 [0.0–18.2] | |
| **Age at first marriage** | | | |
| <20 | 60.2 [58.8–61.6] | 39.8 [38.4–41.2] | <0.0001 |
| 20-24 | 77.3 [74.8–79.6] | 22.7 [20.4–25.2] | |
| 25+ | 76.5 [72.3–80.3] | 23.5 [19.7–27.7] | |
| **Marital duration** | | | |
| 0–4 | 98.6 [97.8–99.1] | 1.5 [0.9–2.2] | <0.0001 |
| 5-9 | 95.2 [93.9–96.3] | 4.8 [3.7–6.1] | |
| 10-19 | 72.2 [69.4–74.8] | 27.8 [25.2–30.6] | |
| 20+ | 31.2 [29.4–33.2] | 68.8 [66.8–70.7] | |
| **Maternal education** | | | |
| No education | 47.7 [44.1–51.2] | 52.3 [48.9–55.9] | <0.0001 |
| Primary | 64.5 [62.9–66.0] | 35.5 [34.0–37.1] | |
| Secondary | 90.9 [90.0–91.7] | 9.1 [8.3–10.0] | |
| Higher | 96.3 [94.4–97.5] | 3.7 [2.5–5.5] | |
| **Wealth index** | | | |
| Poor | 65.9 [64.7–67.2] | 34.1 [32.8–35.3] | <0.0001 |
| Middle | 70.9 [68.8–72.9] | 29.1 [27.1–31.2] | |
| Rich | 86.0 [84.7–87.2] | 14.0 [12.8–15.3] | |
| **Employment status** | | | |
| No | 82.3 [81.2–83.4] | 17.7 [16.6–18.8] | 0.0020 |
| Yes | 68.7 [67.0–70.4] | 31.3 [29.6–33.0] | |
| **Husband/partner education level** | | | |
| None | 46.2 [40.9–51.7] | 53.8 [48.4–59.1] | |
| Primary | 64.7 [63.3 –66.1] | 35.3 [33.9–36.7] | |
| Secondary/higher | 86.4 [83.4–89.0] | 13.6 [11.0–16.7] | |
| **Religion** | | | |
| Catholic | 77.4 [75.4–79.3] | 22.6 [20.7–24.6] | 0.0004 |
| Protestant | 76.1 [75.0–77.1] | 23.9 [22.9–25.0] | |
| Muslim | 47.5 [31.7–63.8] | 52.5 [36.2–68.3] | |
| Other | 77.6 [69.6–84.0] | 22.4 [16.1–30.4] | |
| **Region** | | | |
| Central | 73.7 [70.6–76.6] | 26.3 [23.4–29.4] | <0.0001 |

*(Continued)*

**Table 3.** (Continued)

| Covariates | Zero to low parity (0–4 children) (N=10,423) | Grand multipara (5 or more children) (N=3,260) | (p-value) |
|---|---|---|---|
| | Percent (CI) | Percent (CI) | |
| Copperbelt | 83.4 [80.5–73.0] | 16.6 [14.1–19.4] | |
| Eastern | 70.3 [67.5–73.0] | 29.7 [27.0–32.5] | |
| Luapula | 67.8 [64.8–70.7] | 32.2 [29.3–35.3] | |
| Lusaka | 85.5 [83.6–87.3] | 14.5 [12.7–16.4] | |
| Muchinga | 67.4 [64.2–70.4] | 32.6 [28.1–33.9] | |
| Northern | 69.1 [66.1–71.9] | 30.9 [28.1–33.9] | |
| Northwestern | 76.4 [73.7–9.0] | 23.6 [21.0–26.3] | |
| Southern | 73.2 [69.8–76.3] | 26.8 [23.7–30.2] | |
| Western | 74.3 [71.4–76.9] | 25.7 [23.7–28.6] | |
| **Residence** | | | |
| Urban | 85.3 [83.9–86.6] | 14.7 [13.4–16.1] | <0.0001 |
| Rural | 68.2 [67.0–69.4] | 31.8 [30.6–33.0] | |
| **Access to media** | | | |
| No | 73.5 [72.4–74.6] | 26.5 [25.4–27.7] | <0.0001 |
| Yes | 79.5 [78.1–80.9] | 20.5 [19.1–21.9] | |
| **Marital status** | | | |
| Single | 499.4 [99.1–99.6] | 0.6 [0.4–1.0] | <0.0001 |
| Married | 65.1 [63.8–66.5] | 34.9 [33.6–36.2] | |
| Widowed/Separated/Divorced | 67.8 [65.0–70.4] | 32.2 [29.6–35.0] | |
| **Polygamous marriage** | | | |
| No | 67.5 [66.2–68.7] | 32.5 [31.3–33.8] | <0.0001 |
| Yes | 45.9 [41.7–50.1] | 54.1 [49.9–58.3] | |
| **Contraceptive use** | | | |
| No | 80.1 [78.9–81.2] | 19.9 [18.8–21.1] | 0.0238 |
| Yes | 69.0 [67.6–70.5] | 31.0 [29.5–32.5] | |
| **Previous birth interval** | | | |
| 24 months or more | 60.0 [58.5–61.6] | 40.0 [38.5–41.5] | 0.0067 |
| Less than 24 months | 53.8 [50.0–57.5] | 46.2 [42.5–50.0] | |
| **Access to health care** | | | |
| Big problem | 64.7 [57.5–71.4] | 35.3 [28.6–42.5] | 0.002 |
| Not a big problem | 76.3 [75.3–77.3] | 23.7 [22.7–24.7] | |

p<0.05, CI: Confidence intervals

Women whose previous birth interval was less than 24 months were nearly three times more likely to be grand multiparous compared to women whose previous birth interval was 24 months or longer (aOR=2.74, 95% CI: 1.741–4.326).

## Discussion

This study analyzed over 13,000 women in Zambia and found that approximately one-quarter were grand multipara. Multivariable logistic regression identified advanced maternal age of 35 years or older, early childbearing, rural residency, belonging to a poor wealth quintile, lack of formal education, and longer marital duration of 20 years or more as key risk factors.

The study found that 24% of women in Zambia had experienced grand multiparity. This falls within the range, as most sub-Saharan African (SSA) countries report a prevalence between 17% and 33% [18]. This high prevalence is

**Table 4. Adjusted odds ratios for the binary logistic regression of the association between all independent variables and grand multiparity among adult women aged 15–49 years in Zambia. N = 13683.**

| Covariates | Odds ratios (adjusted) | p-value | (95% CI) |
|---|---|---|---|
| **Age** | | | |
| 15-24 | 1.00 | 0.042 | 1.057–22.355 |
| 25-29 | 4.86 | 0.000 | 3.450–73.645 |
| 30-34 | 15.94 | 0.019 | 1.344–26.087 |
| 35+ | 56.31 | 0.000 | 11.626–272.780 |
| **Age at first birth** | | | |
| <20 | 1.00 | | |
| 20-24 | 0.40 | <0.001 | 0.327–0.507 |
| 25+ | 0.14 | <0.001 | 0.084–0.238 |
| **Age at first marriage** | | | |
| <20 | 1.00 | | |
| 20-24 | 1.06 | 0.737 | 0.766–1.456 |
| 25+ | 1.79 | 0.025 | 1.075–2.969 |
| **Marital duration** | | | |
| 0-4 | 1.00 | | |
| 5-9 | 1.32 | 0.508 | 0.575–3.044 |
| 10-19 | 5.31 | <0.001 | 2.295–12.325 |
| 20+ | 14.10 | <0.001 | 5.854–33.957 |
| **Maternal education** | | | |
| No education | 1.00 | | |
| Primary | 0.82 | 0.179 | 0.611–1.097 |
| Secondary | 0.28 | <0.001 | 0.182–0.418 |
| Higher | 0.09 | <0.001 | 0.046–0.174 |
| **Wealth index** | | | |
| Poor | 1.00 | | |
| Middle | 0.67 | 0.007 | 0.507–0.897 |
| Rich | 0.48 | <0.001 | 0.3337–0.701 |
| **Employment status** | | | |
| No | 1.00 | | |
| Yes | 1.04 | 0.681 | 0.851–1.280 |
| **Region** | | | |
| Central | 1.00 | | |
| Copperbelt | 1.10 | 0.670 | 0.715–1.684 |
| Eastern | 0.78 | 0.194 | 0.540–1.134 |
| Luapula | 1.59 | 0.013 | 1.104–2.305 |
| Lusaka | 0.75 | 0.135 | 0.519–1.093 |
| Muchinga | 1.67 | 0.008 | 1.145–2.439 |
| Northen | 1.27 | 0.272 | 0.825–1.981 |
| Northwestern | 1.35 | 0.185 | 0.865–2.116 |
| Southern | 1.15 | 0.557 | 0.727–1.803 |
| Western | 1.03 | 0.882 | 0.669–1.596 |
| **Partner education** | | | |
| No education | 1.00 | | |
| Primary | 0.87 | 0.377 | 0.638–1.186 |
| Secondary or Higher | 0.80 | 0.428 | 0.476–1.371 |

*(Continued)*

**Table 4.** (Continued)

| Covariates | Odds ratios (adjusted) | p-value | (95% CI) |
|---|---|---|---|
| **Residence** | | | |
| Urban | 1.00 | | |
| Rural | 1.46 | 0.007 | 1.111–1.911 |
| **Contraceptive use** | | | |
| No | 1.00 | | |
| Yes | 1.29 | 0.006 | 1.074–1.545 |
| **Previous birth interval** | | | |
| 24 months or more | 1.00 | | |
| Less than 24 months | 2.74 | <0.001 | 1.741–4.326 |
| **Access to health care** | | | |
| Big problem | 1.00 | | |
| Not a big problem | 0.75 | 0.432 | 0.70–1.531 |

p<0.05, 1.00: Reference Category, aOR: Adjusted Odds Ratio

typically attributed to lower standards of living, inadequate healthcare resources, and limited use of modern contraceptives, compared to low resource settings where prevalence ranges from 3 to 4% [18]. Findings reveal that women aged 35 years or older are at a higher risk for grand multiparity compared to younger women. This is consistent with studies done in Nigeria and Ethiopia [13,21]. This is because older women are more likely to be married and have been exposed to sexual intercourse for a longer period. Maternal health programs should target older women, focusing on safe motherhood and equitable care to address their higher risk of complications like postpartum hemorrhage and pre-eclampsia [22].

Early childbearing was identified as a significant predictor of grand multiparity, with 36% of women who began childbearing before age 20 experiencing grand multiparity, consistent with findings from studies done in SSA [23,24]. Addressing stigma, myths, and negative beliefs through community engagement is key to improving participation in family planning programs [25].

Our study also found that grand multiparity varies with marital duration, women married for over 20 years showed a higher likelihood of grand multiparity, echoing patterns observed in studies from Nigeria and Pakistan [26,27]. This is due to extended reproductive lifespan in long marriages which are likely to contribute to the increased risk [28]. In addition, child marriages can extend marital durations, increasing the likelihood of larger families due to societal norms favoring high fertility in long-term marriages [29]. Despite Zambia's Marriage Act setting the minimum marriage age at 21, exceptions such as parental consent and high court rulings permitting marriages for children under 16 continue to perpetuate child marriage and its associated risks [30].

The study found that women from middle and rich wealth quintile had reduced odds of grand multiparity compared to women from poor wealth quintile. This suggests that socioeconomic status influences reproductive behavior [31], with wealthier women tending to have smaller families. Therefore, addressing this inequality in accessing family planning and reproductive health education could help reduce the risks associated with grand multiparity [1,11,32].

The present study also showed that secondary or higher education attainment were significantly associated with lower odds of grand multiparty. This supports findings from a previous study done in Nigeria [33] that link higher education with increased contraceptive use and smaller family sizes, thereby contributing to reduced maternal mortality [34–36]. Furthermore, educated women are more likely to make informed decisions about family size, underscoring the importance of education in public health strategies [32,37].

The finding of this study revealed that women from rural areas were more likely to experience grand multiparity. This finding indicates that limited access to family planning and healthcare in rural regions contributes to a higher risk of grand multiparity. This disparity highlights the need for targeted interventions to improve reproductive healthcare access in rural areas, such access to family planning services [38]. The study also found that contraceptive use and shorter birth intervals significantly increases the likelihood of grand multiparity. Promoting consistent contraceptive use and healthy birth spacing could reduce the prevalence of grand multiparity and improve maternal and child health outcomes [24,28,32,39]. However, in this study, women who used contraceptives had a higher likelihood of grand multiparity, likely because 56% of our sample were married. In Zambian settings, especially in rural areas, cultural norms encourage larger families, and contraceptive use may begin after several births, leading to higher parity [40]. Inconsistent uptake of contraceptives, particularly oral methods contribute to this trend [41].

About 11% of women in this study were in polygamous unions. Our study did not find polygamous marriage to be a significant predictor of grand multiparity. This aligns with a studies conducted in West Africa and SSA that suggests polygamy does not increase the fertility of women. Nevertheless, women in polygamous marriages have a higher ideal average number of children compared to women in monogamous union as co-wives compete for favor through childbearing and cultural norms encouraging larger families in polygamous unions [42,43].

This study highlights the urgent need for targeted interventions to mitigate risks associated with grand multiparity, particularly among women in rural Zambia, who face elevated risks of postpartum hemorrhage, uterine rupture, and preeclampsia due to factors such as extended marital duration, early childbearing, and low socioeconomic status [44,45]. Strategies should integrate education, family planning, and maternal healthcare services to reduce maternal morbidity and mortality in these high-risk groups [17].

Community-based initiatives, including the deployment of community health workers for education and early risk identification of grand multiparous women, can enhance access to skilled childbirth care. Addressing systemic barriers like transportation, financial limitations, and cultural norms is critical to ensuring equitable healthcare access. Additionally, expanding family planning and education programs to improve birth spacing and reduce unintended pregnancies is essential for achieving improved maternal health outcomes [11,13].

Grand multiparity, remains a public health challenge in Zambia. It is closely linked to higher maternal mortality, poor child health outcomes, and strains on family resources. To address the drivers of grand multiparity, the Zambian government has implemented various policies and strategies targeting maternal health, family planning, and reproductive education. Although these policies do not always explicitly address grand multiparity, they aim to mitigate its underlying causes such as early childbearing, limited access to contraception, and socioeconomic disparities [34–37].

Zambia's National Family Planning focuses on improving access to contraceptives, especially in rural areas where grand multiparity is more common. The strategy, supported by organizations like United Nation Population Fund (UNFPA), aims to reduce unintended pregnancies by making contraceptives widely available [46]. However, gaps in access remain in remote regions, despite improvements in contraceptive prevalence over time [47]. In conjunction with this, the Adolescent Sexual and Reproductive Health (ASRH) Policy targets early pregnancies, a significant factor in grand multiparity [34,48]. By promoting comprehensive sexuality education (CSE) in schools and providing youth-friendly reproductive health services, the ASRH policy seeks to delay the onset of childbearing. Early marriages and pregnancies, particularly in teenage years, often result in larger family sizes, making this policy essential in addressing grand multiparity in Zambia [49].

Zambia's Safe Motherhood Action Plan aims to reduce maternal mortality and improve maternal health, with a key focus on promoting birth spacing. Short birth intervals are a major risk factor for grand multiparity, increasing the likelihood of more pregnancies in a shorter time frame [50]. By advocating for longer intervals between births, the plan seeks to reduce high-risk pregnancies and associated complications, helping to prevent grand multiparity by encouraging fewer pregnancies over a woman's reproductive lifespan [35]. Similarly, the National Health Strategic Plan (2022–2026)

emphasizes improving maternal health and reducing fertility rates [51]. This plan includes expanding family planning services, improving health infrastructure, and promoting female education as a long-term solution to high fertility rates [37]. Education plays a crucial role in reducing grand multiparity, as educated women are more likely to make informed reproductive decisions. The plan targets increased school attendance and educational attainment, particularly in rural areas where fertility rates are highest [27].

While these policies represent a comprehensive approach to addressing the factors contributing to grand multiparity, challenges remain. Zambia still faces significant disparities in healthcare access between rural and urban areas. Women in rural areas often have limited access to family planning services, which increases the likelihood of grand multiparity [25]. Additionally, cultural norms surrounding fertility, particularly in rural regions, sometimes hinder the uptake of contraceptives. Despite these challenges, Zambia's ongoing efforts to expand healthcare access, promote family planning, and improve education are critical steps in reducing the prevalence of grand multiparity.

This study adds to the growing body of literature by stressing the vulnerabilities faced by grand multiparous women in Zambia and SSA at large. The findings provide valuable insights into the policy and programmatic interventions necessary to improve maternal health outcomes for grand multiparous women in Zambia, particularly in rural settings where these women are most disadvantaged. This study also highlights the need for Zambia's maternal health policies to focus on grand multiparous women by improving access to skilled care, addressing barriers like transportation and costs, and enhancing family planning services. Community health workers and awareness campaigns are key to reducing maternal risks and improving outcomes, particularly in rural areas. In addition, studies have shown that high parity women especially those who did not have complications in their last delivery are less likely to deliver at the hospital [52,53]. Further, a study done in Zambia found that high-parity women are more likely to lose their child before they celebrate their first birthday [54].

Zambia has implemented several key reproductive health policies, including the adoption of the International Conference on Population and Development (ICPD) agenda in the early 1990s, which recognized women's sexual and reproductive health rights. In 2005, the government introduced the Reproductive Health Policy to provide free contraceptives in public health facilities, followed by the 2006 Zambia Family Planning National Guidelines, which abolished user fees for family planning services [55]. The government abolished user fees for maternal and child health services to encourage greater utilization of these services by women, particularly in rural areas [56]. Despite these milestones, the impact has been limited. Challenges such as inconsistent contraceptive use, particularly oral methods, religious and cultural barriers, and low education levels among women persist [41]. Moreover, heavy reliance on donor funding has raised concerns about the sustainability and nationwide reach of these interventions [55].

### Strengths and limitations

This study provides valuable insights into grand multiparity among women aged 15–49 years in Zambia using data from a nationally representative Demographic and Health Survey. The use of a large, population-based dataset ensures the representativeness and generalizability of the findings, which strengthens the relevance of the results for national policy and programmatic planning.

However, the study has some limitations. The cross-sectional design limits the ability to draw causal inferences. Additionally, the reliance on self-reported data may introduce recall and social desirability biases. The analysis was also constrained by the variables available in the DHS dataset.

### Conclusion

This study highlights the urgent need to address grand multiparity due to its substantial impact on maternal and child health in Zambia. Key factors associated with grand multiparity include advanced maternal age of 35 years or older, early childbearing lack of formal education and long marital duration of 20 years or more. Additionally, socioeconomic status

plays a significant role, with women from low wealth quintiles generally having more children, while lower educational attainment correlates with a higher prevalence of grand multiparity. Regional disparities further emphasize the need for targeted interventions, as rural women are more likely to experience grand multiparity.

In response to these challenges, a comprehensive policy approach is essential. This includes strengthening provider training on the adverse maternal and child health outcomes of grand multiparity as well as expanding reproductive health education to support informed family planning decisions. Improving the availability of contraceptive and reproductive health services, especially in rural areas, through improved supply chains and outreach programs. Empowering women through education is a key strategy, as higher education levels are associated with improved family planning decisions. Furthermore, promoting longer birth intervals through awareness campaigns and improved healthcare services can help mitigate the risks associated with closely spaced pregnancies. Targeted media campaigns, especially in regions with limited access to information, can play a pivotal role in bridging knowledge gaps and reinforcing the importance of family planning. Future research in Zambia should focus on grand multiparity as a risk factor for adverse maternal and child health outcomes.

## Supporting information

**S1 Table. Multicollinearity analysis of independent variables.**
(DOCX)

## Acknowledgments

We would like to extend our heartfelt thanks to the Zambia Statistics Agency and the DHS Program for allowing us to use the 2018 ZDHS dataset.

## Author contributions

**Conceptualization:** Peter Mumba.

**Data curation:** Peter Mumba.

**Formal analysis:** Peter Mumba, Samson Shumba.

**Methodology:** Peter Mumba, Emmanuel Musonda, Mwangala Mulasinkwanda.

**Writing – original draft:** Peter Mumba, Emmanuel Musonda, Jessy Mutale Nkonde, Vanessa Moonga, Stephen Jere, Samson Shumba.

**Writing – review & editing:** Peter Mumba, Emmanuel Musonda, Mwangala Mulasinkwanda, Jessy Mutale Nkonde, Vanessa Moonga, Stephen Jere, Samson Shumba.

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
