## [Decision Letter · Decision Letter 0]

3 Nov 2024

PGPH-D-24-02228

Grand multiparity and its associated factors in Zambia: Evidence from the 2018 Zambia Demographic and Health Survey.

Dear Dr. Mumba,

Thank you for submitting your manuscript to PLOS Global Public Health. After careful consideration, we feel that it has merit but does not fully meet PLOS Global Public Health’s publication criteria as it currently stands. Therefore, we invite you to submit a revised version of the manuscript that addresses the points raised during the review process.

Please note that we have only been able to secure a single reviewer to assess your manuscript. We are issuing a decision on your manuscript at this point to prevent further delays in the evaluation of your manuscript. Please be aware that the editor who handles your revised manuscript might find it necessary to invite additional reviewers to assess this work once the revised manuscript is submitted. However, we will aim to proceed on the basis of this single review if possible. 

We look forward to receiving your revised manuscript.

Kind regards,

Joanna Tindall, PhD

Staff Editor

Journal Requirements:

1. We have noticed that you have uploaded Supporting Information files, but you have not included a list of legends. Please add a full list of legends for your Supporting Information files after the references list. 

Additional Editor Comments (if provided):

Reviewers' comments:

Reviewer's Responses to Questions

**Comments to the Author**

1. Does this manuscript meet PLOS Global Public Health’s publication criteria? Is the manuscript technically sound, and do the data support the conclusions? The manuscript must describe methodologically and ethically rigorous research with conclusions that are appropriately drawn based on the data presented.

Reviewer #1: Partly

2. Has the statistical analysis been performed appropriately and rigorously?

Reviewer #1: No

3. Have the authors made all data underlying the findings in their manuscript fully available (please refer to the Data Availability Statement at the start of the manuscript PDF file)?

Reviewer #1: No

4. Is the manuscript presented in an intelligible fashion and written in standard English?

Reviewer #1: Yes

5. Review Comments to the Author

Reviewer #1: The authors describe the prevalence of grand multiparity in a sample of Zambian women with a recent birth, and the associated factors.

Grand multiparity is an important but often overlooked risk factor, common in sub-Saharan Africa. It is an important analysis, as grand multiparity has mostly been studied in hospitals, thus on a subgroup of high parity women who reach hospitals for childbirth. Population-based studies are not common. I have read the analysis with interest.

Major comments -

1. The authors describe and discuss the prevalence of grand multiparity. In view that a part of the women were excluded - women under 20 and women with 0 or 1 children - the study cannot report prevalence in the country, but only in their sample. Thus the study cannot report that 39% of women in Zambia had experienced grand multiparity. 39% of women in this sample were grand multiparous. If all women had been included in the denominator, the study could report the percentage of grand multiparity in the population. As it is, researchers must report that this is the percentage in their sample. The percentage reported overestimates the percentage of grand multiparous women in the population in Zambia.

I feel that the study would improve if the researcher reported characteristics of all women (thus reporting the prevalence in the population) at the beginning, then describing analysis of the subgroup they have chosen to analyse.

Additionally, I think women who are parity over 10 should not be excluded from the "grand multiparity" group. It is a small number, and will not change analysis significantly. However, they share the same risk factors.

2. The significance of the paper will increase if the authors examine the significance of the risk factor from a public health point of view. Specifically, the discussion will benefit by considering the public health significance of this sizable group. The authors have focused on prevention of the condition (education, FP). However, the problem of high fertility in sub-Saharan is not easy to solve, and these women and their babies exist and are at higher risk. Additionally, morbidity and mortality among these women have wide impact among their families and communities. This was described by Fathalla in "The long road to maternal death", in 1987. What can be done in communities, facilities for these women? What are the implication of this research in current maternal health care policy in Zambia?

Also, should comment on how their findings fit into the existing literature on women of high parity in sub-Saharan Africa. Recent multicountry evidence (including Zambia in the analysis) (https://www.ncbi.nlm.nih.gov/pubmed/38262683) indicates that GM (or high parity) is a vulnerability that adds on to poverty in rural settings. Women of high parity rarely give birth in hospitals, despite being at increased risk. For example, how does the study add to the picture of disadvantage of these women?

Minor comments:

Reference 4 - The correct name of the author is Salomon

In the introduction, authors should note that Few studies have examined grand multiparity from a population perspective. Most studies are based on hospital populations

Low-resource/high resource settings are preferable terms compared to developed/developing

Methods

Authors should explain why variable wealth has been categorized into poor, middle, rich, rather than into quintiles

Title Table 2 – Bivariate analysis of women’s parity (low parity and grand multiparity) by background characteristics

Results

“Maternal education was strongly associated with grand multiparity”. This sentence should be reworded to maternal education showed a strong negative association with GM

Discussion - see above

6. PLOS authors have the option to publish the peer review history of their article (what does this mean?). If published, this will include your full peer review and any attached files.

**Do you want your identity to be public for this peer review?** For information about this choice, including consent withdrawal, please see our Privacy Policy.

Reviewer #1: No

---

## [Decision Letter · Decision Letter 1]

14 Feb 2025

PGPH-D-24-02228R1

Grand multiparity and its associated factors in Zambia: Evidence from the 2018 Zambia Demographic and Health Survey.

Dear Mumba,

Thank you for submitting your manuscript to PLOS Global Public Health. After careful consideration, we feel that it has merit but does not fully meet PLOS Global Public Health’s publication criteria as it currently stands. Therefore, we invite you to submit a revised version of the manuscript that addresses the points raised during the review process.

We look forward to receiving your revised manuscript.

Kind regards,

Collins Otieno Asweto, PhD

Academic Editor

Journal Requirements:

Reviewer's Responses to Questions

**Comments to the Author**

1. If the authors have adequately addressed your comments raised in a previous round of review and you feel that this manuscript is now acceptable for publication, you may indicate that here to bypass the “Comments to the Author” section, enter your conflict of interest statement in the “Confidential to Editor” section, and submit your "Accept" recommendation.

Reviewer #1: (No Response)

Reviewer #2: (No Response)

2. Does this manuscript meet PLOS Global Public Health’s publication criteria? Is the manuscript technically sound, and do the data support the conclusions? The manuscript must describe methodologically and ethically rigorous research with conclusions that are appropriately drawn based on the data presented.

Reviewer #1: Yes

Reviewer #2: Partly

3. Has the statistical analysis been performed appropriately and rigorously?

Reviewer #1: Yes

Reviewer #2: Yes

4. Have the authors made all data underlying the findings in their manuscript fully available (please refer to the Data Availability Statement at the start of the manuscript PDF file)?

Reviewer #1: Yes

Reviewer #2: No

5. Is the manuscript presented in an intelligible fashion and written in standard English?

Reviewer #1: Yes

Reviewer #2: No

6. Review Comments to the Author

Reviewer #1: PGPH-D-24-02228R1

Overall, the authors have addressed the concerns contained in the previous review. The manuscript has gained in clarity and now findings are derived from the statistical analysis.

I recommend the following minor changes:

1. Abstract

First sentence

There are socio-economic determinants to high fertility, ( poverty and low women’s education), rather than just lack of access to contraception/medical care. The sentence should be reworded to ( …) high resources setting, partly due to …

2. Methods

Third paragraph/page 4

Text refers to Table 5, but I cannot find it.

Also, authors report that there was no collinearity between independent variables, but then that marital status was dropped due to multicollinearity.

Please address this.

3. Results

Table 2 – 1 decimal for percentages is sufficient

Table 4 – as the authors have correctly reported p-values, asterisks are not necessary and should be removed to make the table more easily readable.

4. Discussion – I suggest adding a first paragraph summarizing your main findings.

Eg this study analysed over 13000 women to study prevalence and risk factors associated with grand multiparity in Zambia. We found approximately one quarter of women in Zambia were grand multipara. Using multivariate logistic regression, we found age, length of marriage duration, poverty, etc were associated to grand multiparity.

The discussion of the findings should then follow.

5. Conclusion – I suggest to move the conclusion to the end. It should summarize findings and relevant policy recommendations. The first sentence reporting risks associated to grand multiparity from existing literature should not be part of the conclusion.

Reviewer #2: -In fact, ZDHS 2024 was published. I could not find any indication to publish this article that analyses ZDHS 2018 data.

7. PLOS authors have the option to publish the peer review history of their article (what does this mean?). If published, this will include your full peer review and any attached files.

**Do you want your identity to be public for this peer review?** For information about this choice, including consent withdrawal, please see our Privacy Policy.

Reviewer #1: No

Reviewer #2: **Yes: **Dr. Khursheda Akhtar

---

## [Decision Letter · Decision Letter 2]

31 Mar 2025

PGPH-D-24-02228R2

Grand multiparity and its associated factors in Zambia: Evidence from the 2018 Zambia Demographic and Health Survey.

Dear Mumba,

Thank you for submitting your manuscript to PLOS Global Public Health. After careful consideration, we feel that it has merit but does not fully meet PLOS Global Public Health’s publication criteria as it currently stands. Therefore, we invite you to submit a revised version of the manuscript that addresses the points raised during the review process.

We look forward to receiving your revised manuscript.

Kind regards,

Collins Otieno Asweto, PhD

Academic Editor

Journal Requirements:

Additional Editor Comments (if provided):

Reviewers' comments:

Reviewer's Responses to Questions

**Comments to the Author**

1. If the authors have adequately addressed your comments raised in a previous round of review and you feel that this manuscript is now acceptable for publication, you may indicate that here to bypass the “Comments to the Author” section, enter your conflict of interest statement in the “Confidential to Editor” section, and submit your "Accept" recommendation.

Reviewer #3: All comments have been addressed

Reviewer #4: All comments have been addressed

2. Does this manuscript meet PLOS Global Public Health’s publication criteria? Is the manuscript technically sound, and do the data support the conclusions? The manuscript must describe methodologically and ethically rigorous research with conclusions that are appropriately drawn based on the data presented.

Reviewer #3: Partly

Reviewer #4: Yes

3. Has the statistical analysis been performed appropriately and rigorously?

Reviewer #3: No

Reviewer #4: Yes

4. Have the authors made all data underlying the findings in their manuscript fully available (please refer to the Data Availability Statement at the start of the manuscript PDF file)?

Reviewer #3: Yes

Reviewer #4: Yes

5. Is the manuscript presented in an intelligible fashion and written in standard English?

Reviewer #3: No

Reviewer #4: Yes

6. Review Comments to the Author

Reviewer #3: Generally you have no new things added, by default your significant factors are known and familiar, you can get from different obstetrics books, so what new thigs you add rather than known factors? This concept disagree with goal of research. It was better, if you were study the prevalence with maternal as well as fetal outcomes of grand multiparity.

Reviewer #4: The paper is interesting and provides valuable insights into the prevalence and related factors of grand multiparity in Zambia. It highlights a critical topic that should be properly addressed to reduce high fertility rates mainly in low-income countries. In general, the document is rich and well structured. Yet, the below comments would help improve its quality.

Abstract:

It is not easy to read mainly for the statistic part. Simplify and provide clear sentences.

Background:

This part is comprehensive, but it is important to provide some statistics concerning the prevalence of grand multiparity.

Provide some details about the context and the way it may affect the fertility rate.

Highlight the initiatives taken by the instances to reduce fertility rate. Ok I found this explanation in the discussion unless you want to reflect on it briefly in the background.

Methods

I did not understand why you excluded women of less than 20. In sub-Saharan African, including Zambia, women may become sexually active at a very early age and are more exposed to multiparity. This is revealed just at the beginning of the results where you mentioned that “more than two-thirds (68%) of these women had their first childbirth before the age of 20, and the majority (91%) initiated sexual activity before turning 20.”

p. 8: Age at first birth <20, 20-34, 25+. 34 should be 24.

Accessibility to healthcare services is a confounding variable that is not included in the table and ought to be mentioned in the limitations unless you add it in the analysis.

Results

What percentage of women were in the younger age groups (15-19, 20-24)?

Arrange the % in table 2

Polygamous marriage (number of other wives)

No 4532(67.46%) 2186(32.54%)

Yes 389(45.89%) 459(55411%)

Does partners’ education affect grand parity? Why this was excluded from the analysis?

Discussion

Women who used contraceptives were 29% more likely to experience grand multiparity compared to non-users. This could be interpreted the way you did but it might be that women are not using the contraceptive methods properly because of low education and poor life conditions with no access to media information. They might also be using contraceptive methods that are not so reliable.

How do you explain the below results:

Women with a previous pregnancy interval of 24 months or more had more than twice the odds of grand multiparity compared to those with shorter intervals (aOR=2.76, 95% CI: 1.764-4.321).

This requires explanation and justification, as usually the opposite is true.

Discuss the potential influence of polygamy on reproductive patterns as this is present among 11% of the participants.

It might be also interesting to further discuss why the implemented policies did not properly meet their aims. This could be related to the healthcare infrastructure, limited number and unskillful human resources, accessibility and adaptability of service, lack of culturally sensitive services, etc.

Conclusion

Add study limitations.

Recommendations to improve reproductive health services to reduce grand multiparity focusing on the gaps in healthcare services and the need to equip healthcare providers to address this issue more efficiently.

Edit the paper.

7. PLOS authors have the option to publish the peer review history of their article (what does this mean?). If published, this will include your full peer review and any attached files.

**Do you want your identity to be public for this peer review?** For information about this choice, including consent withdrawal, please see our Privacy Policy.

Reviewer #3: No

Reviewer #4: **Yes: **Mathilde Azar

---

## [Decision Letter · Decision Letter 3]

28 Jul 2025

Grand multiparity and its associated factors in Zambia: Evidence from the 2018 Zambia Demographic and Health Survey.

PGPH-D-24-02228R3

Dear Mr Mumba,

We are pleased to inform you that your manuscript 'Grand multiparity and its associated factors in Zambia: Evidence from the 2018 Zambia Demographic and Health Survey.' has been provisionally accepted for publication in PLOS Global Public Health.

Best regards,

Julia Robinson

Executive Editor

Reviewer Comments (if any, and for reference):

Reviewer's Responses to Questions

**Comments to the Author**

1. If the authors have adequately addressed your comments raised in a previous round of review and you feel that this manuscript is now acceptable for publication, you may indicate that here to bypass the “Comments to the Author” section, enter your conflict of interest statement in the “Confidential to Editor” section, and submit your "Accept" recommendation.

Reviewer #5: All comments have been addressed

2. Does this manuscript meet PLOS Global Public Health’s publication criteria? Is the manuscript technically sound, and do the data support the conclusions? The manuscript must describe methodologically and ethically rigorous research with conclusions that are appropriately drawn based on the data presented.

Reviewer #5: Yes

3. Has the statistical analysis been performed appropriately and rigorously?

Reviewer #5: Yes

4. Have the authors made all data underlying the findings in their manuscript fully available (please refer to the Data Availability Statement at the start of the manuscript PDF file)?

Reviewer #5: Yes

5. Is the manuscript presented in an intelligible fashion and written in standard English?

Reviewer #5: Yes

6. Review Comments to the Author

Reviewer #5: This is a well and carefully written manuscript. The authors have demonstrated sufficient knowledge of the study. However some of the few minor revisions are listed below:

1. Table 3 "Northen" instead of Northern, remove the backlash after "Protestant", check cell spacing and ensure consistency.

2. Table 4 for 25-29 the OR reads "4,86" rather than 4.86.

7. PLOS authors have the option to publish the peer review history of their article (what does this mean?). If published, this will include your full peer review and any attached files.

**Do you want your identity to be public for this peer review?** For information about this choice, including consent withdrawal, please see our Privacy Policy.

Reviewer #5: **Yes: **Queen Esther Adeyemo
